# GRAPH HYPERNETWORKS FOR NEURAL ARCHITECTURE SEARCH

**Chris Zhang**[1,2]**, Mengye Ren**[1,3] **& Raquel Urtasun**[1,3]
[1]Uber Advanced Technologies Group, [2]University of Waterloo, [3]University of Toronto
`cjzhang@edu.uwaterloo.ca, {mren3,urtasun}@uber.com`

## ABSTRACT

Neural architecture search (NAS) automatically finds the best task-specific neural network topology, outperforming many manual architecture designs. However, it can be prohibitively expensive as the search requires training thousands of different networks, while each can last for hours. In this work, we propose the Graph HyperNetwork (GHN) to amortize the search cost: given an architecture, it directly generates the weights by running inference on a graph neural network. GHNs model the topology of an architecture and therefore can predict network performance more accurately than regular hypernetworks and premature early stopping. To perform NAS, we randomly sample architectures and use the validation accuracy of networks with GHN generated weights as the surrogate search signal. GHNs are fast – they can search nearly $10\times$ faster than other random search methods on CIFAR-10 and ImageNet. GHNs can be further extended to the anytime prediction setting, where they have found networks with better speed-accuracy tradeoff than the state-of-the-art manual designs.

## 1 INTRODUCTION

The success of deep learning marks the transition from manual feature engineering to automated feature learning. However, designing effective neural network architectures requires expert domain knowledge and repetitive trial and error. Recently, there has been a surge of interest in *neural architecture search* (NAS), where neural network architectures are automatically optimized.

One approach for architecture search is to consider it as a nested optimization problem, where the inner loop finds the optimal parameters $w^*$ for a given architecture $a$ w.r.t. the training loss $\mathcal{L}_{train}$, and the outer loop searches the optimal architecture w.r.t. a validation loss $\mathcal{L}_{val}$:

$$w^*(a) = \arg\min_{w} \mathcal{L}_{train}(w, a) \tag{1}$$

$$a^* = \arg\min_{a} \mathcal{L}_{val}(w^*(a), a) \tag{2}$$

Traditional NAS is expensive since solving the inner optimization in Eq. 1 requires a lengthy optimization process (e.g. stochastic gradient descent (SGD)). Instead, we propose to learn a parametric function approximation referred to as a hypernetwork (Ha et al., 2017; Brock et al., 2018), which attempts to *generate* the network weights directly. Learning a hypernetwork is an amortization of the cost of solving Eq. 1 repeatedly for multiple architectures. A trained hypernetwork is well correlated with SGD and can act as a much faster substitute.

Yet, the architecture of the hypernet itself is still to be determined. Existing methods have explored a variety of tactics to represent architectures, such as an ingenious 3D tensor encoding scheme (Brock et al., 2018), or a string serialization processed by an LSTM (Zoph & Le, 2017; Zoph et al., 2018; Pham et al., 2018). In this work, we advocate for a *computation graph* representation as it allows for the topology of an architecture to be explicitly modeled. Furthermore, it is intuitive to understand and can be easily extensible to various graph sizes.

To this end, in this paper we propose the *Graph HyperNetwork* (GHN), which can aggregate graph level information by directly learning on the graph representation. Using a hypernetwork to guide architecture search, our approach requires significantly less computation when compared to state-of-the-art methods. The computation graph representation allows GHNs to be the first hypernetwork to generate all the weights of arbitrary CNNs rather than a subset (e.g. Brock et al. (2018)), achieving stronger correlation and thus making the search more efficient and accurate.

While the validation accuracy is often the primary goal in architecture search, networks must also be resource aware in real-world applications. Towards this goal, we exploit the flexibility of the GHN by extending it to the problem of *anytime prediction*. Models capable of anytime prediction progressively update their predictions, allowing for a prediction at any time. This is desirable in settings such as real-time systems, where the computational budget available for each test case may vary greatly and cannot be known ahead of time. Although anytime models have non-trivial differences to classical models, we show the GHN is amenable to these changes.

We summarize our main contributions of this work:

1. We propose Graph HyperNetwork that predicts the parameters of unseen neural networks by directly operating on their computational graph representations.

2. Our approach achieves highly competitive results with state-of-the-art NAS methods on both CIFAR-10 and ImageNet-mobile and is $10\times$ faster than other random search methods.

3. We demonstrate that our approach can be generalized and applied in the domain of anytime-prediction, previously unexplored by NAS programs, outperforming the existing manually designed state-of-the-art models.

## 2 RELATED WORK

Various search methods such as reinforcement learning (Zoph & Le, 2017; Baker et al., 2017a; Zoph et al., 2018), evolutionary methods (Real et al., 2017; Miikkulainen et al., 2017; Xie & Yuille, 2017; Liu et al., 2018b; Real et al., 2018) and gradient-based methods (Liu et al., 2018c; Luo et al., 2018) have been proposed to address the outer optimization (Eq. 2) of NAS, where an agent learns to sample architectures that are more likely to achieve higher accuracy. Different from these methods, this paper places its focus on the inner-loop: inferring the parameters of a given network (Eq. 1). Following Brock et al. (2018); Bender et al. (2018), we opt for a simple random search algorithm to complete the outer loop.

While initial NAS methods simply train candidate architectures for a brief period with SGD to obtain the search signal, recent approaches have proposed alternatives in the interest of computational cost. Baker et al. (2017b) propose directly predicting performance from the learning curve, and Deng et al. (2017) propose to predict performance directly from the architecture without learning curve information. However, training a performance predictor requires a ground truth, thus the expensive process of computing the inner optimization is not avoided. Pham et al. (2018); Bender et al. (2018); Liu et al. (2018c) use parameter sharing, where a "one-shot" model containing all possible architectures in the search space is trained. Individual architectures are sampled by deactivating some nodes or edges in the one-shot model. In this case, predicting $w^*(a)$ can be seen as using a selection function from the set of parameters in the one-shot model.

Prior work has shown the feasibility of predicting parameters in a network with a function approximator (Denil et al., 2013). Schmidhuber (1992; 1993) proposed "fast-weights", where one network produces weight changes for another. HyperNetworks (Ha et al., 2017) generate the weights of another network and show strong results in large-scale language modeling and image classification experiments. SMASH (Brock et al., 2018) applied HyperNetworks to perform NAS, where an architecture is encoded as a 3D tensor using a memory channel scheme. In contrast, we encode a network as a computation graph and use a graph neural network. While SMASH predicts a subset of the weights, our graph model is able to predict *all* the free weights.

While earlier NAS methods focused on standard image classification and language modeling, recent literature has extended NAS to search for architectures that are computationally efficient (Tan et al., 2018; Dong et al., 2018; Hsu et al., 2018; Elsken et al., 2018; Zhou et al., 2018). In this work, we applied our GHN based search program on the task of anytime prediction, where we not only optimize for the final speed but the entire speed-accuracy trade-off curve.

## 3 BACKGROUND

We review the two major building blocks of our model: graph neural networks and hypernetworks.

**Graph Neural Network:** A graph neural network (Scarselli et al., 2009; Li et al., 2016; Kipf & Welling, 2017) is a collection of nodes and edges $(\mathcal{V}, \mathcal{E})$, where each node is a recurrent neural network (RNN) that individually sends and receives messages along the edges, spanning over the

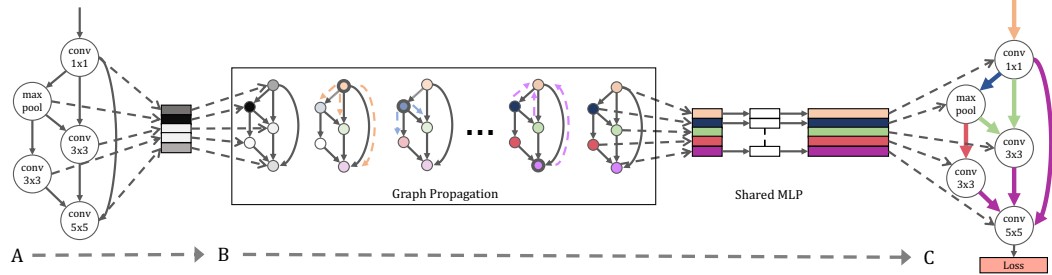

Figure 1: Our system diagram. **A**: A neural network architecture is randomly sampled, forming a GHN. **B**: After graph propagation, each node in the GHN generates its own weight parameters. **C**: The GHN is trained to minimize the training loss of the sampled network with the generated weights. Random networks are ranked according to their performance using GHN generated weights.

horizon of message passing. Each node $v$ stores an internal node embedding vector $\boldsymbol{h}_v^{(t)} \in \mathbb{R}^D$, and is updated recurrently:

$$\boldsymbol{h}_v^{(t+1)} = \begin{cases} U\left(\boldsymbol{h}_v^{(t)}, \boldsymbol{m}_v^{(t)}\right) & \text{if node } v \text{ is active,} \\ \boldsymbol{h}_v^{(t)} & \text{otherwise,} \end{cases} \tag{3}$$

where $U$ is a recurrent cell function and $\boldsymbol{m}_v^{(t)}$ is the message received by $v$ at time step $t$:

$$\boldsymbol{m}_v^{(t)} = \sum_{u \in N_{in}(v)} M\left(\boldsymbol{h}_u^{(t)}\right), \tag{4}$$

with $M$ the message function and $N_{in}(v)$ the set of neighbors with incoming edges pointing towards $v$. $U$ is often modeled with a long short-term memory (LSTM) unit (Hochreiter & Schmidhuber, 1997) or gated recurrent unit (GRU) (Cho et al., 2014), and $M$ with an MLP. Given a graph $\mathcal{A}$, we define the GNN operator $G_\mathcal{A}$ to be a mapping from a set of initial node embeddings $\{\boldsymbol{h}_v^{(0)}\}$ to a set of different node embeddings $\{\boldsymbol{h}_v^{(t)}\}$, parameterized by some learnable parameters $\boldsymbol{\phi}$:

$$\left\{\boldsymbol{h}_v^{(t)} | v \in \mathcal{V}\right\} = G_\mathcal{A}^{(t)}\left(\left\{\boldsymbol{h}_v^{(0)} | v \in \mathcal{V}\right\}; \boldsymbol{\phi}\right). \tag{5}$$

Throughout propagation the node embeddings $\boldsymbol{h}_v^{(t)}$ continuously aggregate graph level information, which can be used for tasks such as node prediction and graph prediction by further aggregation. Similar to RNNs, GNNs are typically learned using backpropagation through time (BPTT) (Werbos, 1990).

**Hypernetwork:**  A hypernetwork (Ha et al., 2017) is a neural network that generates the parameters of another network. For a typical deep feedforward network with $D$ layers, the parameters of the $j$-th layer $W_j$ can be generated by a learned function $H$:

$$W_j = H(z_j), \quad \forall j = 1, \ldots, D, \tag{6}$$

where $z_j$ is the layer embedding, and $H$ is shared for all layers. The output dimensionality of the hypernetwork is fixed, but it's possible to accommodate predicting weights for layers of varying kernel sizes by concatenating multiple kernels of the fixed size. Varying spatial sizes can also be accommodated by slicing in the spatial dimensions. Hypernetworks have been found effective in standard image recognition and text classification problems, and can be viewed as a relaxed weight sharing mechanism. Recently, they have shown to be effective in accelerating architecture search (Brock et al., 2018).

## 4 GRAPH HYPERNETWORKS FOR NEURAL ARCHITECTURAL SEARCH

Our proposed Graph HyperNetwork (GHN) is a composition of a graph neural network and a hypernetwork. It takes in a computation graph (CG) and generates all free parameters in the graph. During evaluation, the generated parameters are used to evaluate the fitness of a random architecture, and the top performer architecture on a separate validation set is then selected. This allows us to search over a large number of architectures at the cost of training a single GHN. We refer the reader to Figure 1 for a high level system overview.

### 4.1 GRAPHICAL REPRESENTATION

We represent a given architecture as a directed acyclic graph $\mathcal{A} = (\mathcal{V}, \mathcal{E})$, where each node $v \in \mathcal{V}$ has an associated computational operator $f_v$ parametrized by $w_v$, which produces an output activation tensor $x_v$. Edges $e_{u \mapsto v} = (u, v) \in \mathcal{E}$ represent the flow of activation tensors from node $u$ to node $v$. $x_v$ is computed by applying its associated computational operator on each of its inputs and taking summation as follows

$$x_v = \sum_{e_{u \mapsto v} \in \mathcal{E}} f_v(x_u; w_v), \quad \forall v \in \mathcal{V}. \tag{7}$$

### 4.2 GRAPH HYPERNETWORK

Our proposed Graph Hypernetwork is defined as a composition of a GNN and a hypernetwork. First, given an input architecture, we used the graphical representation discussed above to form a graph $\mathcal{A}$. A parallel GNN $G_{\mathcal{A}}$ is then constructed to be *homomorphic* to $\mathcal{A}$ with the exact same topology. Node embeddings are initialized to one-hot vectors representing the node's computational operator. After graph message-passing steps, a hypernet uses the node embeddings to generate each node's associated parameters. Let $\boldsymbol{h}_v^{(T)}$ be the embedding of node $v$ after $T$ steps of GNN propagation, and let $H(\cdot; \boldsymbol{\varphi})$ be a hypernetwork parametrized by $\boldsymbol{\varphi}$, the generated parameters $\tilde{\boldsymbol{w}}_v$ are:

$$\tilde{\boldsymbol{w}}_v = H\left(\boldsymbol{h}_v^{(T)}; \boldsymbol{\varphi}\right). \tag{8}$$

For simplicity, we implement $H$ with a multilayer perceptron (MLP). It is important to note that $H$ is shared across all nodes, which can be viewed as an output prediction branch in each node of the GNN. Thus the final set of generated weights of the entire architecture $\tilde{\boldsymbol{w}}$ is found by applying $H$ on all the nodes and their respective embeddings which are computed by $G_{\mathcal{A}}$:

$$\tilde{\boldsymbol{w}} = \{\tilde{\boldsymbol{w}}_v | \, v \in \mathcal{V}\} = \left\{ H\left(\boldsymbol{h}_v^{(T)}; \boldsymbol{\varphi}\right) \mid v \in \mathcal{V} \right\} \tag{9}$$

$$= \left\{ H\left(\boldsymbol{h}; \boldsymbol{\varphi}\right) \mid \boldsymbol{h} \in G_{\mathcal{A}}^{(T)}\left(\left\{\boldsymbol{h}_v^{(0)} | v \in \mathcal{V}\right\}; \boldsymbol{\phi}\right) \right\} \tag{10}$$

$$= GHN\left(\mathcal{A}; \boldsymbol{\phi}, \boldsymbol{\varphi}\right). \tag{11}$$

### 4.3 ARCHITECTURAL MOTIFS AND STACKED GNNs

The computation graph of some popular CNN architectures often spans over hundreds of nodes (He et al., 2016a; Huang et al., 2017), which makes the search problem scale poorly. Repeated architecture motifs are originally exploited in those architectures where the computation of each computation block at different resolutions is the same, e.g. ResNet (He et al., 2016b). Recently, the use of architectural motifs also became popular in the context of neural architecture search, e.g. (Zoph et al., 2018; Pham et al., 2018), where a small graph module with a fewer number of computation nodes is searched, and the final architecture is formed by repeatedly stacking the same module. Zoph et al. (2018) showed that this leads to stronger performance due to a reduced search space; the module can also be transferred to larger datasets by adopting a different repeating pattern.

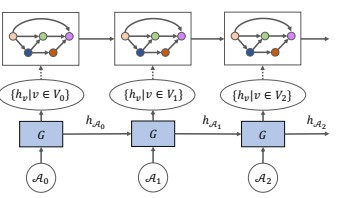

Figure 2: Stacked GHN along the depth dimension.

Our proposed method scales naturally with the design of repeated modules by stacking the same graph hypernetwork along the depth dimension. Let $\mathcal{A}$ be a graph composed of a chain of repeated modules $\{\mathcal{A}_i\}_{i=1}^N$. A graph level embedding $\boldsymbol{h}_{\mathcal{A}_i}$ is computed by taking an average over all node embeddings after a full propagation of the current module, and passed onwards to the input node of the next module as a message before graph propagation continues to the next module.

$$\boldsymbol{h}_{\mathcal{A}_0} = 0, \tag{12}$$

$$\boldsymbol{h}_{\mathcal{A}_i} = \frac{1}{|\mathcal{V}_i|} \sum_{v \in \mathcal{V}_i} \left\{\boldsymbol{h}_v^{(T)} | v \in \mathcal{V}_i\right\} \tag{13}$$

$$= \frac{1}{|\mathcal{V}_i|} \sum G_{\mathcal{A}_i}^{(T)}\left(\left\{\boldsymbol{h}_v^{(0)} | v \in \mathcal{V}_i\right\}, \boldsymbol{h}_{\mathcal{A}_{i-1}}; \boldsymbol{\phi}\right) \quad \forall i > 0 \tag{14}$$

Note that $G_{\mathcal{A}_i}$ share parameters for all $\mathcal{A}_i$. Please see Figure 2 for an overview.

Table 1: Comparison with image classifiers found by state-of-the-art NAS methods which employ a random search on CIFAR-10. Results shown are mean $\pm$ standard deviation.

| Method | Search Cost (GPU days) | Param $\times 10^6$ | Accuracy |
|---|---|---|---|
| SMASHv1 (Brock et al., 2018) | ? | 4.6 | 94.5 |
| SMASHv2 (Brock et al., 2018) | 3 | 16.0 | 96.0 |
| One-Shot Top (F=32) (Bender et al., 2018) | 4 | $2.7 \pm 0.3$ | $95.5 \pm 0.1$ |
| One-Shot Top (F=64) (Bender et al., 2018) | 4 | $10.4 \pm 1.0$ | $95.9 \pm 0.2$ |
| Random (F=32) | - | $4.6 \pm 0.6$ | $94.6 \pm 0.3$ |
| GHN Top (F=32) | 0.42 | $5.1 \pm 0.6$ | $95.7 \pm 0.1$ |

## 4.4 FORWARD-BACKWARD GNN MESSAGE PASSING

Standard GNNs employ the *synchronous propagation scheme* (Li et al., 2016), where the node embeddings of all nodes are updated simultaneously at every step (see Equation 3). Recently, Liao et al. (2018) found that such propagation scheme is inefficient in passing long-range messages and suffers from the vanishing gradient problem as do regular RNNs. To mitigate these shortcomings they proposed *asynchronous propagation* using graph partitions. In our application domain, deep neural architectures are chain-like graphs with a long diameter; This can make synchronous message passing difficult. Inspired by the backpropagation algorithm, we propose another variant of asynchronous propagation scheme, which we called *forward-backward* propagation, that directly mimics the order of node execution in a backpropagation algorithm. Specifically, let $s$ be a topological sort of the nodes in the computation graph in a forward pass,

$$
\boldsymbol{h}_v^{(t+1)} = \begin{cases} U\left(\boldsymbol{h}_v^{(t)}, \boldsymbol{m}_v^{(t)}\right) & \text{if } s(t) = v \text{ and } 1 \leq t \leq |\mathcal{V}| \\ & \text{or if } s(2|\mathcal{V}| - t) = v \text{ and } |\mathcal{V}| + 1 \leq t < 2|\mathcal{V}|, \\ \boldsymbol{h}_v^{(t)} & \text{otherwise.} \end{cases} \tag{15}
$$

The total number of propagation steps $T$ for a full forward-backward pass will then become $2|\mathcal{V}| - 1$. Under the synchronous scheme, propagating information across a graph with diameter $|\mathcal{V}|$ would require $O(|\mathcal{V}|^2)$ messages. This is reduced to $O(|\mathcal{V}|)$ under the forward-backward scheme.

## 4.5 LEARNING

Learning a graph hypernetwork is straightforward since $\tilde{\boldsymbol{w}}$ are directly generated by a differentiable network. We compute gradients of the graph hypernetwork parameters $\phi, \varphi$ using the chain rule:

$$
\nabla_{\phi,\varphi} \mathcal{L}_{train}(\tilde{\boldsymbol{w}}) = \nabla_{\tilde{\boldsymbol{w}}} \mathcal{L}_{train}(\tilde{\boldsymbol{w}}) \cdot \nabla_{\phi,\varphi} \tilde{\boldsymbol{w}} \tag{16}
$$

The first term is the gradients of standard network parameters, the second term is decomposed as

$$
\nabla_{\phi} \tilde{\boldsymbol{w}} = \left\{ \nabla_{\boldsymbol{h}} H(\boldsymbol{h}; \varphi) \cdot \nabla_{\phi} \boldsymbol{h} \mid \boldsymbol{h} \in G^{(T)}\left(\{\boldsymbol{h}_v^{(0)}\}, \mathcal{A}, \phi\right) \right\}, \tag{17}
$$

$$
\nabla_{\varphi} \tilde{\boldsymbol{w}} = \left\{ \nabla_{\varphi} H(\boldsymbol{h}_v^{(T)}; \varphi) \mid v \in \mathcal{V} \right\} \tag{18}
$$

where (Eq. 17) is the contribution from GNN module $G$ and (Eq. 18) is the contribution from the hypernet module $H$. Both $G$ and $H$ are jointly learned throughout training.

## 5 EXPERIMENTS

In this section, we use our proposed GHN to search for the best CNN architecture for image classification. First, we evaluate the GHN on the standard CIFAR (Krizhevsky & Hinton, 2009) and ImageNet (Russakovsky et al., 2015) architecture search benchmarks. Next, we apply GHN on an "anytime prediction" task where we optimize the speed-accuracy tradeoff that is key for many real-time applications. Finally, we benchmark the GHN's predicted-performance correlation and explore various factors in an ablation study.

## 5.1 NAS BENCHMARKS

### 5.1.1 CIFAR-10

We conduct our initial set of experiments on CIFAR-10 (Krizhevsky & Hinton, 2009), which contains 10 object classes and 50,000 training images and 10,000 test images of size $32\times32\times3$. We use 5,000 images split from the training set as our validation set.

Table 2: Comparison with image classifiers found by state-of-the-art NAS methods which employ advanced search methods on CIFAR-10. Results shown are mean $\pm$ standard deviation.

| Method | Search Cost (GPU days) | Param $\times 10^6$ | Accuracy |
|---|---|---|---|
| NASNet-A (Zoph et al., 2018) | 1800 | 3.3 | 97.35 |
| ENAS Cell search (Pham et al., 2018) | 0.45 | 4.6 | 97.11 |
| DARTS (first order) (Liu et al., 2018c) | 1.5 | 2.9 | 97.06 |
| DARTS (second order) (Liu et al., 2018c) | 4 | 3.4 | $97.17 \pm 0.06$ |
| GHN Top-Best, 1K (F=32) | 0.84 | 5.7 | $97.16 \pm 0.07$ |

Table 3: Comparison with image classifiers found by state-of-the-art NAS methods which employ advanced search methods on ImageNet-Mobile.

| Method | Search Cost (GPU days) | Param $\times 10^6$ | FLOPs $\times 10^6$ | Accuracy Top 1 | Top 5 |
|---|---|---|---|---|---|
| NASNet-A (Zoph et al., 2018) | 1800 | 5.3 | 564 | 74.0 | 91.6 |
| NASNet-C (Zoph et al., 2018) | 1800 | 4.9 | 558 | 72.5 | 91.0 |
| AmoebaNet-A (Real et al., 2018) | 3150 | 5.1 | 555 | 74.5 | 92.0 |
| AmoebaNet-C (Real et al., 2018) | 3150 | 6.4 | 570 | 75.7 | 92.4 |
| PNAS (Liu et al., 2018a) | 225 | 5.1 | 588 | 74.2 | 91.9 |
| DARTS (second order) (Liu et al., 2018c) | 4 | 4.9 | 595 | 73.1 | 91.0 |
| GHN Top-Best, 1K | 0.84 | 6.1 | 569 | 73.0 | 91.3 |

**Search space:** Following existing NAS methods, we choose to search for optimal blocks rather than the entire network. Each block contains 17 nodes, with 8 possible operations. The final architecture is formed by stacking 18 blocks. The spatial size is halved and the number of channels is doubled after blocks 6 and 12. These settings are all chosen following recent NAS methods (Zoph & Le, 2017; Pham et al., 2018; Liu et al., 2018c), with details in the Appendix.

**Training:** For the GNN module, we use a standard GRU cell (Cho et al., 2014) with hidden size 32 and 2 layer MLP with hidden size 32 as the recurrent cell function $U$ and message function $M$ respectively. The shared hypernetwork $H(\cdot; \varphi)$ is a 2-layer MLP with hidden size 64. From the results of ablations studies in Section 5.4, the GHN is trained with blocks with $N = 7$ nodes and $T = 5$ propagations under the forward-backward scheme, using the ADAM optimizer (Kingma & Ba, 2015). Training details of the final selected architectures are chosen to follow existing works and can be found in the Appendix.

**Evaluation:** First, we compare to similar methods that use random search with a hypernetwork or a one-shot model as a surrogate search signal. We randomly sample 10 architectures and train until convergence for our random baseline. Next, we randomly sample 1000 architectures, and select the top 10 performing architectures with GHN generated weights, which we refer to as GHN Top. Our reported search cost includes both the GHN training and evaluation phase. Shown in Table 1, the GHN achieves competitive results with nearly an order of magnitude reduction in search cost.

In Table 2, we compare with methods which use more advanced search methods, such as reinforcement learning and evolution. Once again, we sample 1000 architectures and use the GHN to select the top 10. To make a fair comparison for random search, we train the top 10 for a short period before selecting the best to train until convergence. The accuracy reported for GHN Top-Best is the average of 5 runs of the same final architecture. Note that all methods in Table 2 use CutOut (Devries & Taylor, 2017). GHN achieves very competitive results with a simple random search algorithm, while only using a fraction of the total search cost. Using advanced search methods with GHNs may bring further gains.

### 5.1.2 IMAGENET-MOBILE

We also run our GHN algorithm on the ImageNet dataset (Russakovsky et al., 2015), which contains 1.28 million training images. We report the top-1 accuracy on the 50,000 validation images. Following existing literature, we conduct the ImageNet experiments in the mobile setting, where the model is constrained to be under 600M FLOPS. We directly transfer the best architecture block found in the CIFAR-10 experiments, using an initial convolution layer of stride 2 before stacking 14 blocks with scale reduction at blocks 1, 2, 6 and 10. The total number of flops is constrained by choosing the initial number of channels. We follow existing NAS methods on the training procedure of the final architecture; details can be found in the Appendix. As shown in Table 3 the transferred block is competitive with other NAS methods which require a far greater search cost.

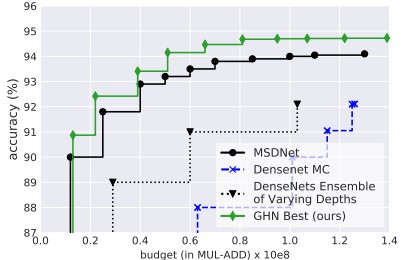

Figure 3: Comparison with state-of-the-art human-designed networks on CIFAR-10.

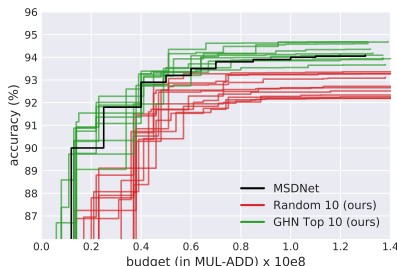

Figure 4: Comparison between random 10 and top 10 networks on CIFAR-10.

Table 4: Benchmarking the correlation between the predicted and true performance of the GHN against SGD and a one-shot model baselines. Results are on CIFAR-10.

| Method | Computation cost | | Correlation | |
|---|---|---|---|---|
| | Initial (GPU hours) | Per arch. (GPU seconds) | Random-100 | Top-50 |
| SGD 10 Steps | - | 0.9 | 0.26 | -0.05 |
| SGD 100 Steps | - | 9 | 0.59 | 0.06 |
| SGD 200 Steps | - | 18 | 0.62 | 0.20 |
| SGD 1000 Steps | - | 90 | 0.77 | 0.26 |
| One-Shot | 9.8 | 0.06 | 0.58 | 0.31 |
| GHN | 6.1 | 0.08 | 0.68 | 0.48 |

## 5.2 ANYTIME PREDICTION

In the real-time setting, the computational budget available can vary for each test case and cannot be known ahead of time. This is formalized in anytime prediction, (Grubb & Bagnell, 2012) the setting in which for each test example $\mathbf{x}$, there is non-deterministic computational budget $B$ drawn from the joint distribution $P(\mathbf{x}, B)$. The goal is then to minimize the expected loss $L(f) = \mathbb{E}\left[L\left(f(\mathbf{x}), B\right)\right]_{P(\mathbf{x}, B)}$, where $f(\cdot)$ is the model and $L(\cdot)$ is the loss for an $f(\cdot)$ that must produce a prediction within the budget $B$.

We conduct experiments on CIFAR-10. Our anytime search space consists of networks with 3 cells containing 24, 16, and 8 nodes. Each node is given the additional properties: 1) the spatial size it operates at and 2) if an early-exit classifier is attached to it. A node enforces its spatial size by pooling or upsampling any input feature maps inputs that are of different scale. Note that while a naive one-shot model would triple its size to include three different parameter sets at three different scales, the GHN is negligibly affected by such a change. The GHN uses the area under the predicted accuracy-FLOPS curve as its selection criteria. The search space, contains various convolution and pooling operators. Training methodology of the final architectures are chosen to match Huang et al. (2018) and can be found in the Appendix.

Figure 3 shows a comparison with the various methods presented by Huang et al. (2018). Our experiments show that the best searched architectures can outperform the current state-of-the-art human designed networks. We see the GHN is amenable to the changes proposed above, and can find efficient architectures with a random search when used with a strong search space.

## 5.3 PREDICTED PERFORMANCE CORRELATION (CIFAR-10)

In this section, we evaluate whether the parameters generated from GHN can be indicative of the final performance. Our metric is the correlation between the accuracy of a model with trained weights vs. GHN generated weights. We use a fixed set of 100 random architectures that have not been seen by the GHN during training, and we train them for 50 epochs to obtain our "ground-truth" accuracy, and finally compare with the accuracy obtained from GHN generated weights. We report the Pearson's R score on all 100 random architectures and the top 50 performing architectures (i.e. above average architectures). Since we are interested in searching for the best architecture, obtaining a higher correlation on top performing architectures is more meaningful.

To evaluate the effectiveness of GHN, we further consider two baselines: 1) training a network with SGD from scratch for a varying number of steps, and 2) our own implementation of the one-

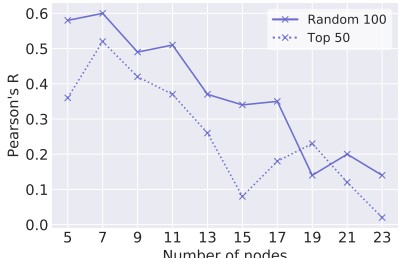
(a) Vary number of nodes; $T = 5$, forward-backward

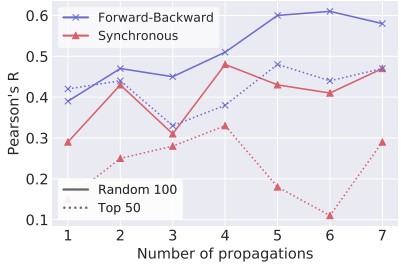
(b) Vary propagation schemes, $N = 7$

Figure 5: GHN when varying the number of nodes and propagation scheme

shot model proposed by Pham et al. (2018), where nodes store a set of shared parameters for each possible operation. Unlike GHN, which is compatible with varying number of nodes, the one-shot model must be trained with $N = 17$ nodes to match the evaluation. The GHN is trained with $N = 7, T = 5$ using forward-backward propagation. These GHN parameters are selected based on the results found in Section 5.4.

Table 4 shows performance correlation and search cost of SGD, the one-shot model, and our GHN. Note that GHN clearly outperforms the one-shot model, showing the effectiveness of dynamically predicting parameters based on graph topology. While it takes 1000 SGD steps to surpasses GHN in the "Random-100" setting, GHN is still the strongest in the "Top-50" setting, which is more important for architecture search. Moreover, compared to GHN, running 1000 SGD steps for every random architecture is over 1000 times more computationally expensive. In contrast, GHN only requires a pre-training stage of 6 hours, and afterwards, the trained GHN can be used to efficiently evaluate a massive number of random architectures of different sizes.

## 5.4 Ablation Studies (CIFAR-10)

**Number of graph nodes:** The GHN is compatible with varying number of nodes - graphs used in training need not be the same size as the graphs used for evaluation. Figure 5a shows how GHN performance varies as a function of the number of nodes employed during training - fewer nodes generally produces better performance. While the GHN has difficulty learning on larger graphs, likely due to the vanishing gradient problem, it can generalize well from just learning on smaller graphs. Note that all GHNs are tested with the full graph size ($N = 17$ nodes).

**Number of propagation steps:** We now compare the forward-backward propagation scheme with the regular synchronous propagation scheme. Note that $T = 1$ synchronous step corresponds to one full forward-backward phase. As shown in Figure 5b, the forward-backward scheme consistently outperforms the synchronous scheme. More propagation steps also help improving the performance, with a diminishing return. While the forward-backward scheme is less amenable to acceleration from parallelization due to its sequential nature, it is possible to parallelize the evaluation phase across multiple GHNs when testing the fitness of candidate architectures.

**Stacked GHN for architectural motifs:** We also evaluate different design choices of GHNs on representing architectural motifs. We compare 1) individual GHNs, each predicting one block independently, 2) a stacked GHN where individual GHN's pass on their graph embedding without sharing parameters, 3) a stacked GHN with shared parameters (our proposed approach). As shown in Table 5, passing messages between GHN's is crucial, and sharing parameters produces better performance.

| SP | PE | Correlation | |
| --- | --- | --- | --- |
| | | Random-100 | Top-50 |
| ✗ | ✗ | 0.24 | 0.15 |
| ✗ | ✓ | 0.44 | 0.37 |
| ✓ | ✓ | 0.68 | 0.48 |

Table 5: Stacked GHN Correlation. SP denotes sharing parameters and PE denotes passing embeddings

## 6 Conclusion

In this work, we propose the Graph HyperNetwork (GHN), a composition of graph neural networks and hypernetworks that generates the weights of any architecture by operating directly on their computation graph representation. We demonstrate a strong correlation between the performance with the generated weights and the fully-trained weights. Using our GHN to form a surrogate search signal, we achieve competitive results on CIFAR-10 and ImageNet mobile with nearly $10\times$ faster speed compared to other random search methods. Furthermore, we show that our proposed method can be extended to outperform the best human-designed architectures in setting of anytime prediction, greatly reducing the computation cost of real-time neural networks.

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

## 7 APPENDIX

### 7.1 SEARCH SPACE

**Standard image classification on CIFAR-10 and ImageNet**  The search space for CIFAR-10 and ImageNet classification experiments includes the following operations:

- identity
- $1 \times 1$ convolution
- $3 \times 3$ separable convolution
- $5 \times 5$ separable convolution
- $3 \times 3$ dilated separable convolution
- $5 \times 5$ dilated separable convolution
- $1 \times 7$ convolution followed $7 \times 1$ convolution
- $3 \times 3$ max pooling
- $3 \times 3$ average pooling

A block forms an output by concatenating all leaf nodes in the graph. Blocks have 2 input nodes which ingest the output of block $i - 1$ and block $i - 2$ respectively. The input nodes are bottleneck layers, and can reduce the spatial size by using stride 2.

Note that while ENAS supports only 5 operators due to memory constraints, GHNs can search for more operators. This is because ENAS (and other methods which use one-shot models) must store all the parameters in memory because it finds paths in a larger model. Thus the memory requirements are $O(KN)$ where $K$ is the number of operations and $N$ is the number of nodes in the candidate architecture. In contrast, the memory requirement for GHNs is $O(N) + O(K)$ for the candidate architecture and GHN respectively.

**Anytime prediction on CIFAR-10**  The search space for the CIFAR-10 anytime prediction experiments includes the following operations:

- $1 \times 1$ convolution
- $3 \times 3$ convolution
- $5 \times 5$ convolution
- $3 \times 3$ max pooling
- $3 \times 3$ average pooling

In the anytime setting, nodes concatenate their inputs rather than sum. Thus, the identity operator was removed as it would be redundant. The search space does not include separable convolutions so that it is comparable with our baselines (Huang et al., 2018). Block 1 contains nodes which may operate on any of the 3 scales ($32 \times 32, 16 \times 16, 8 \times 8$). Block 2 contains nodes which can only operate on scales $16 \times 16$ and $8 \times 8$. Block 3 only contains nodes which operate on the scale $8 \times 8$. We fix the number of exit nodes. These choices are inspired by Huang et al. (2018)

### 7.2 GRAPH HYPERNETWORK DETAILS

**Standard image classification on CIFAR-10 and ImageNet**  While node embeddings are initialized to a one-hot vector representing computational operator of the node, we found it helpful to pass the sparse vector through a learned embedding matrix prior to graph propagation. The GHN is trained for 200 epochs with batch size 64 using the ADAM optimizer with an initial learning rate 1e-3 that is divided by 2 at epoch 100 and 150. A naive hypernet would have a separate output branch for each possible node type, and simply ignore branches that aren't applicable to the specific node. In this manner, the number of parameters of the hypernetwork scale according to the number of possible node computations. In contrast, the number of parameters for a one-shot model scale according to the number of nodes in the graph. We further reduce number of parameters by obtaining smaller sized convolutions kernels through the slicing of larger sized kernels.

**Anytime prediction** In the anytime prediction setting, two one-hot vectors representing the node's scale and presence of an early exit classifier are additionally concatenated to the first initialized node embedding. We found it helpful to train the GHN with a random number of nodes per block, with maximum number of allowed nodes being the evaluation block size. Because nodes concatenate their inputs, a bottleneck layer is required. The hypernetwork can predict bottleneck parameters for a varying number of input nodes by generating weights based on edge activations rather than node activations. We form edge activations by concatenating the node activations of the parent and child. Edge weights generated this way can be concatenated, allowing the dimensionality of the predicted bottleneck weights the be proportional to the number of incoming edges.

## 7.3 FINAL ARCHITECTURE TRAINING DETAILS

**CIFAR-10** Following existing NAS methods (Zoph et al., 2018; Real et al., 2018), the final candidates are trained for 600 epochs using SGD with momentum 0.9, a single period cosine schedule with $l_{max} = 0.025$, and batch size 64. For regularization, we use scheduled drop-path with a final dropout probability of 0.4. We use an auxiliary head located at 2/3 of the network weighted by 0.5. We accelerate training by performing distributed training across 32 GPUs; the learning rate is multiplied by 32 with an initial linear warmup of 5 epochs.

**ImageNet Mobile** For ImageNet mobile experiments, we use an image size of $224 \times 224$. Following existing NAS methods (Zoph et al., 2018; Real et al., 2018), the final candidates are trained for 250 epochs using SGD with momentum 0.9, initial learning rate 0.1 multiplied by 0.97 every epoch. We use an auxiliary head located at 2/3 of the network weighted by 0.5. We use the same regularization techniques, and similarly accelerate training in a distributed fashion.

**Anytime** Following Huang et al. (2018), the final candidates are trained using SGD with momentum 0.9. We train the models for 300 epochs use an initial learning rate of 0.1, which is divided by 10 after 150 and 225 epochs using a batch size of 64. We accelerate training with distributed training in a similar fashion as the CIFAR-10 classification and ImageNet mobile experiments. The number of filters for the final architecture is chosen such that the number of FLOPS is comparable to existing baselines.

## 7.4 INVESTIGATING ACCURACY DROP OFF

Figure 6 shows a plot comparing the accuracy of an architecture that is trained for 50 epochs and the accuracy of the same architecture using GHN generated weights.

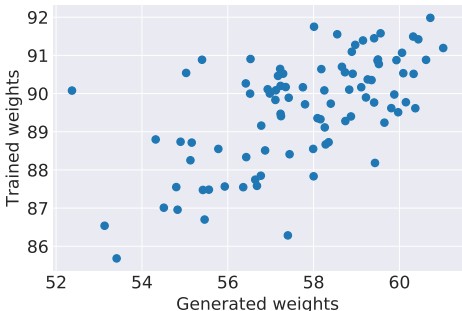

Figure 6: Comparison for 100 randomly sampled architectures.

## 7.5 VISUALIZATION OF FINAL ARCHITECTURES

### 7.5.1 CIFAR-10 AND IMAGENET CLASSIFICATION

Figure 7 shows the best found block in the CIFAR-10 Experiments.

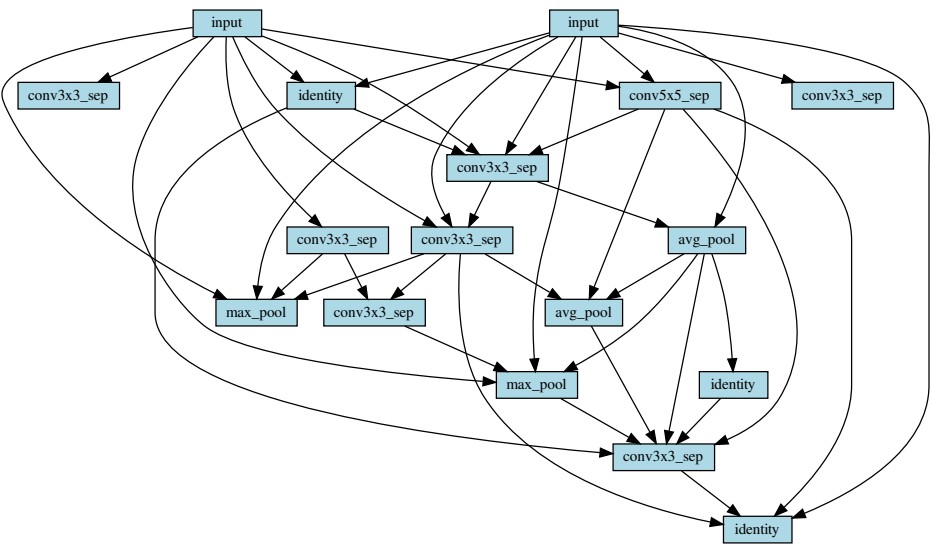

Figure 7: Best block found for classification

### 7.5.2 ANYTIME PREDICTION

Figures 8, 9 and 10 show blocks 1 2 and 3 of the best architecture found in the anytime experiments. The color red denotes that an early exit is attached to the output of the node.

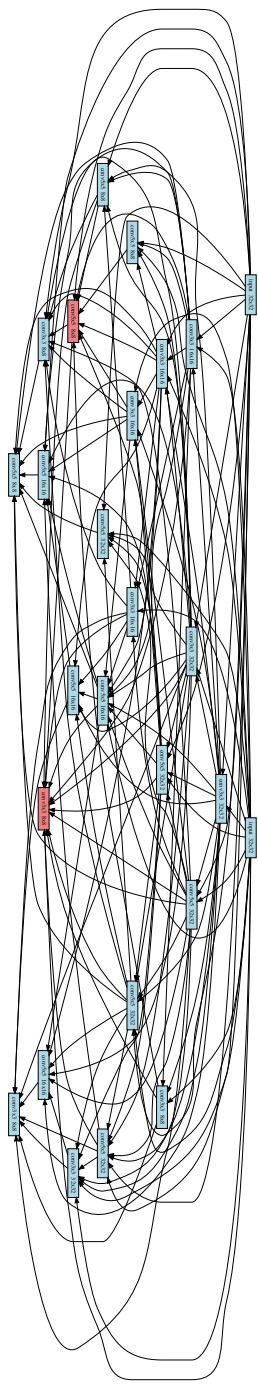

Figure 8: Block 1 for anytime network. Red color denotes early exit.

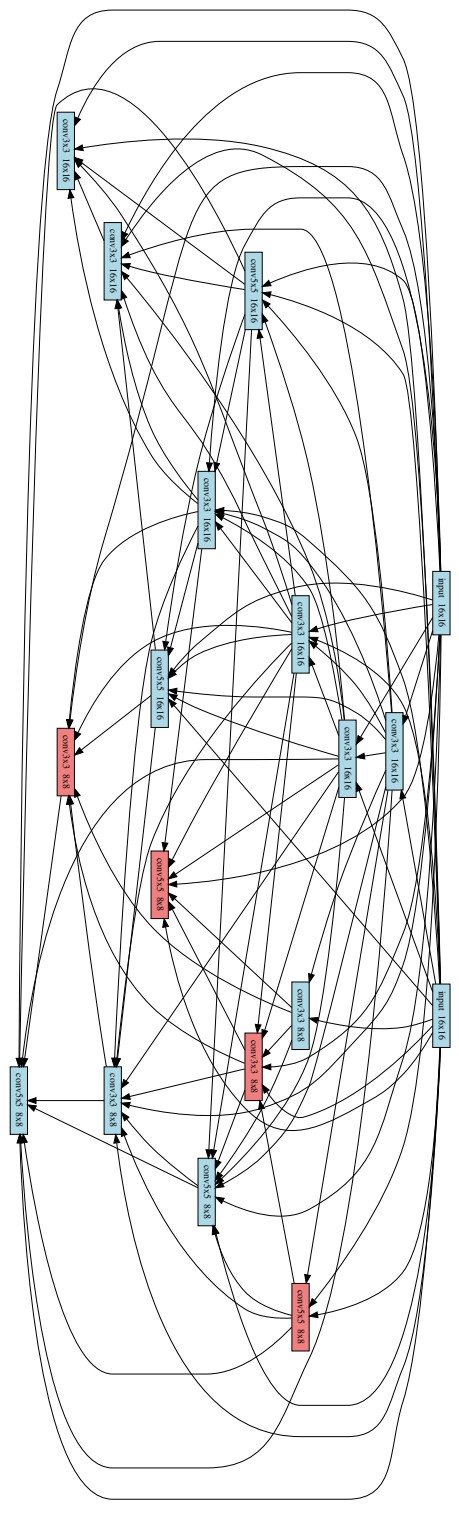

Figure 9: Block 2 for anytime network. Red color denotes early exit.

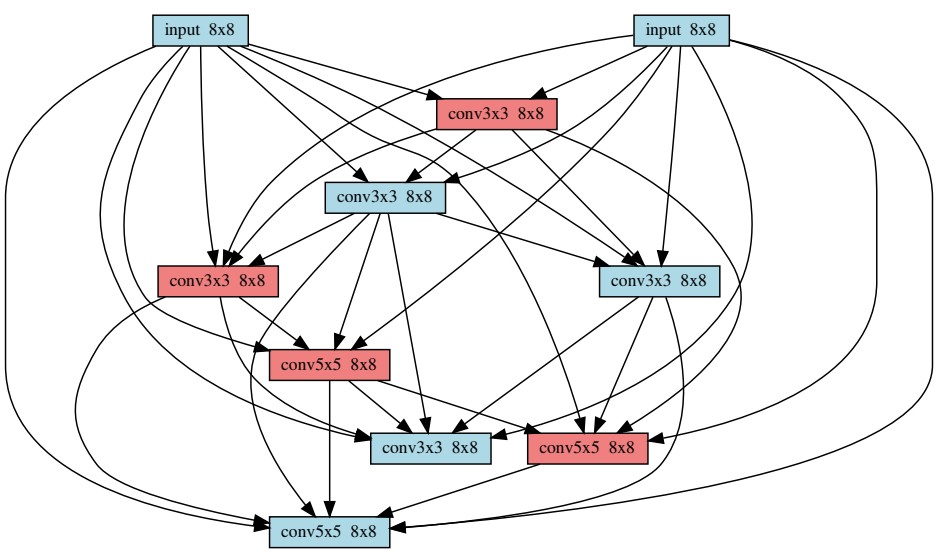

Figure 10: Block 3 for anytime network. Red color denotes early exit.

