# OpenReview forum: "Graph HyperNetworks for Neural Architecture Search"
_ICLR.cc/2019/Conference_

### Official Review · AnonReviewer1 · 2018-10-27
**Review 1 for "Graph HyperNetworks for Neural Architecture Search"**

**Rating:** 7
**Confidence:** 4

**Review:**

This paper proposes to accelerate architecture search by replacing the expensive inner loop (wherein candidate architectures are trained to completion) with a HyperNetwork which predicts the weights of candidate architectures, as in SMASH. Contrary to SMASH, this work employs a Graph neural network to allow for the use of any feedforward architecture, enabling fast architecture search through parameter prediction using highly performant search spaces. The authors test their system and show that performance using Graph HyperNet-generated weights correlates with performance when trained normally. The authors benchmark their method against competing approaches ("traditional" NAS techniques which incur the full expense of the inner loop, and one-shot techniques which learn a large model then select architectures by searching for paths in said model) and show competitive performance.

This is a solid technical contribution with a well-designed set of experiments. While the novelty is not especially high, the paper does a good job of synthesizing existing tools and achieves reasonably strong results with much less compute, making for a strong entry into the growing table of fast architecture search methods. I argue in favor of acceptance.

Notes:

-Whereas SMASH is limited to architectures which can be described with its proposed encoding scheme, GHNs only requires that the architecture be represented as a graph (which, to my knowledge, means it can handle any feedforward architecture).

-Section 4.2: It's not entirely clear how this setup allows for variable sized kernels or variable #channels. Is the output of H simply as large as the largest allowable parameter tensor, and sliced as necessary? A snippet of code might be more illuminating here than a set of equations. Additionally (I may have missed this in the text) is the #channels in each node held fixed with a predfined pattern, or also searched for? Are the channels for each node within a block allowed to vary relative to one another?

-Do you sample a new, random architecture at every SGD step during training of the GHN?

-I have no expertise in graph neural networks, and I cannot judge the efficacy of this scheme wrt other GNN techniques, nor can I judge the forward-backward message passing scheme of section 4.4. If another reviewer has expertise in this area and can provide an evaluation that would be great.

-GPU-days is an okay metric, but it's also problematic, since it will of course depend on the choice of GPU (e.g. you can achieve a 10x speedup just from switching from a 600-series to a V100! How does using 4 GPUS for 1 hour compare to 1 GPU for 4 hours? How does this change if you have more CPU power and can load data faster? What if you're using a DL framework which is faster than your competitor's?) Given that the difference here is an order of magnitude, I don't think it matters, but if authors begin to optimize for GPU-milliseconds then it will need to be better standardized.

-Further empirical evidence showing the correlation between approximate performance and true performance is also strong. I very much like that this study has been run for a method based on finding paths in a larger model (ENAS) and shows that ENAS' performance does indeed correlate with true performance, *but* not perfectly, something which (if I recall correctly) is not addressed in the original paper.

-It is worth noting that for ImageNet-Mobile and CIFAR-10 they perform on par with the top methods but tend to use more parameters.

-I like figures 3 and 4, the comparisons against MSDNet and random networks as a function of op budget is good to see.

-Table 4 shows that the correlation is weaker (regardless of method) for the top architectures, which I don't find surprising as I would expect the variation in performance amongst top architectures to be lower. It would be interesting to also see what the range of error rates are; I would expect that the correlation is higher when the range of error rates across the population of architectures is large, as it is easier to distinguish very bad architectures from very good architectures. Distinguishing among a set of good-to-very-good architectures is likely to be more difficult.

-For Section 5.3, I found the choice to use unseen architectures a little bit confusing. I think that even for this study, there's no reason to use a held-out set, as we seek to scrutinize the ability of the system to approximate performance even with architectures it *does* see during training.

-How much does the accuracy drop when using GHN weights? I would like to see a plot showing true accuracy vs. accuracy with GHN weights for the random-100 networks, as using approximations like this typically results in the approximated weights being substantially worse. I am curious to see just how much of a drop there is.

-Section 5.4: it's interesting that performance is stronger when the GHN only sees a few (7) nodes during training, even though it sees 17 nodes during testing. I would expect that the best performance is attained with training-testing parity. Again, as I do not have any expertise in graph neural networks, I'm not sure if this is common (to train on smaller graphs and generalize to larger ones), so if the authors or another reviewer would like to comment and further illuminate this behavior, that would be helpful.

Some typos:

Abstract: "prematured"  should be "premature"

Introducton, last paragraph: "CNN networks." CNN already stands for Convolutional Neural Network.

---

> ### Author Response · Authors · 2018-11-27
> **Response to Reviewer 1**
>
> We thank the reviewer for their evaluation! To answer the questions:
>
> >> “Section 4.2: It's not entirely clear how this setup allows for variable sized kernels or variable #channels … is the #channels in each node held fixed with a predefined pattern, or also searched for? Are the channels for each node within a block allowed to vary relative to one another?”
>
> Yes, the output of H is as large as the largest parameter tensor and sliced as necessary. The number of channels is held fixed with a predefined pattern (doubling after each reduction). They are not searched for and do not vary relative to one another
>
> >> “Do you sample a new, random architecture at every SGD step during training of the GHN?”
>
> Yes, a new, random architecture is sampled at every SGD step during training of the GHN
>
> >> “GPU-days is an okay metric, but it's also problematic, since it will of course depend on the choice of GPU (e.g. you can achieve a 10x speedup just from switching from a 600-series to a V100! How does using 4 GPUS for 1 hour compare to 1 GPU for 4 hours? How does this change if you have more CPU power and can load data faster? What if you're using a DL framework which is faster than your competitor's?) Given that the difference here is an order of magnitude, I don't think it matters, but if authors begin to optimize for GPU-milliseconds then it will need to be better standardized.”
>
> Yes, we agree that a standardized metric may be necessary as GPU timings become lower and lower. To be clear, for our experiments, we use a single GTX 1080Ti with PyTorch. Additionally, we don’t find data-loading to be a bottleneck for CIFAR-10.
>
> >>”For Section 5.3, I found the choice to use unseen architectures a little bit confusing. I think that even for this study, there's no reason to use a held-out set, as we seek to scrutinize the ability of the system to approximate performance even with architectures it *does* see during training. ”
>
> We initially used a repeated held-out set to save computation during earlier experiments. Note that in practice due to the size of the search space, no architecture is seen twice anyways. However, an interesting avenue for future work would be investigating a hypernetwork’s ability to ‘overfit’ to architectures.
>
> >> “How much does the accuracy drop when using GHN weights? I would like to see a plot showing true accuracy vs. accuracy with GHN weights for the random-100 networks, as using approximations like this typically results in the approximated weights being substantially worse. I am curious to see just how much of a drop there is.”
>
> Regarding accuracy dropoff:  Please see the updated appendix with plots comparing  accuracy with generated weights vs. trained weights
>
> >>”Section 5.4: it's interesting that performance is stronger when the GHN only sees a few (7) nodes during training, even though it sees 17 nodes during testing. I would expect that the best performance is attained with training-testing parity. Again, as I do not have any expertise in graph neural networks, I'm not sure if this is common (to train on smaller graphs and generalize to larger ones), so if the authors or another reviewer would like to comment and further illuminate this behavior, that would be helpful.”
>
> We suspect that the GHN has difficulty learning due to the vanishing gradients when passing messages across large graphs. We believe that the forward-backward passing scheme partially addresses this as it reduces the total number of messages passed. Exploring additional methods to help the GHN learn on larger graphs is an interesting avenue for future work.

---

### Official Review · AnonReviewer2 · 2018-11-02
**Combing Graph Neural Networks and Hyper Networks for NAS**

**Rating:** 6
**Confidence:** 4

**Review:**

This paper proposes using graph neural network (GNN) as hypernetworks to generate free weight parameters for arbitrary CNN architectures. The achieved performance is satisfactory (e.g., error rate < 3 on CIFAR-10 with cutout). I’m particularly interested in the results on ImageNet: it seems the discovered arch on CIFAR-10 (with less than 1 GPU day) successfully transferred to ImageNet.

Generally speaking, the paper is comprehensive in studying the effects of GNN acting as hypernetworks for NAS.  The idea is clear, and the experiments are satisfactory. There are no technical flaws per my reading. The writing is also easy to follow.
On the other hand, the extension of using GNN is indeed natural and straightforward compared with (Brock et al. 2018). Towards that end, the contribution and novelty of the paper is largely marginal and not impressive.

Question:
1.	The authors mention that ‘the first hypernetwork to generate all the weights of arbitrary CNN networks rather than a subset (Brock et al. 2018)’. I’m sorry that I do not understand the particular meaning of such a statement, especially given the only difference of this work with (Brock et al. 2018) lies in how to represent NN architectures. I am not clear that why encoding via 3D tensor cannot “generate all weights”, but can only generate only “a subset”. Furthermore, I’m very curious about the effectiveness of representing the graph using LSTM encoding, and then feeding it to the hypernetworks, since simple LSTM encoding is shown to be very powerful [1]. This at least, should act as a baseline.

2.	Can the authors give more insights about why they can search on 9 operators within less than 1 GPU day? I mean that for example ENAS, can only support 5 operators due to GPU memory limitation (on single GPU card). Do the authors use more than one GPU to support the search process?
Finally, given the literature of NAS is suffering from the issue of reproduction, I do hope the authors could release their codes and detailed pipelines.

[1] Luo, Renqian, et al. "Neural architecture optimization." NIPS (2018).

---

> ### Author Response · Authors · 2018-11-27
> **Response to Reviewer 2**
>
> We thank the reviewer for their evaluation!
>
> To answer the questions:
>
> >> “The authors mention that ‘the first hypernetwork to generate all the weights of arbitrary CNN networks rather than a subset (Brock et al. 2018)’. I’m sorry that I do not understand the particular meaning of such a statement, especially given the only difference of this work with (Brock et al. 2018) lies in how to represent NN architectures. I am not clear that why encoding via 3D tensor cannot “generate all weights”, but can only generate only “a subset”.
>
> The SMASH encoding method is formulated such that it generates weights only for the 1x1 convolution bottleneck layers. While it certainly may be possible for a to augment SMASH or propose a new 3D tensor encoding method to generate all weights, we are not aware of such a method yet. However, the graph representation lends itself to straightforwardly generate all weights.
>
>
> >> “Furthermore, I’m very curious about the effectiveness of representing the graph using LSTM encoding, and then feeding it to the hypernetworks, since simple LSTM encoding is shown to be very powerful [1]. This at least, should act as a baseline”
>
> Unfortunately, we have not run an LSTM-Hypernet baseline, and are not aware of any existing methods, and we agree this would be interesting future work. However, we do compare with ENAS, which uses a weight sharing mechanism and an LSTM encoding with a controller.  As Reviewer 2 has pointed out, [1] has shown very strong results with an LSTM controller and a continuous optimization method. However, the graph method does carry some distinct advantages. For example, as Reviewer 1 pointed out, the graph representation is flexible enough to handle a varying number of neighbours (where the number of neighbours has been conventionally fixed for LSTM representations).
>
> >> “Can the authors give more insights about why they can search on 9 operators within less than 1 GPU day? I mean that for example ENAS, can only support 5 operators due to GPU memory limitation (on single GPU card). Do the authors use more than one GPU to support the search process? “
>
> For ENAS, it must store all the parameters in memory because it finds paths in a larger model. Thus the memory requirements are O (KN) where K is the number of operations and N is the number of nodes in the candidate architecture. In contrast, the memory requirement for GHNs is O(N) + O(K) for the candidate architecture and GHN respectively. Thus, memory is not an issue, and we conduct GHN training on a single GTX 1080Ti.

---

> > ### Comment · AnonReviewer2 · 2018-12-05
> > **Thanks for the response**
> >
> > Thanks for the further explanation especially on the memory usage. I'm fine with this part.  However, the authors seem not adequatly addressed the other two concerns.
> >
> > First, for the LSTM encoding baseline, I'm not quite sure about the validness of "the number of neighbours has been conventionally fixed for LSTM representations" since we can always perform traversal on graph to form a sequence. Even though the authors are right, I'm not convinced about the importance to "handle a varying number of neighbours" in NAS, since there are no empirical evidences supporting that.
> >
> > Second, the authors have not mentioned anything about code publish/reproducibility. I do agree with the public comment that *the reproduction of code is an essential step to make a solid NAS paper*,  otherwise the community has nothing except yet another paper/(unfair) baseline. I have to be quite conservative to recommend an acceptance if I'm not guaranteed that the experimental results could be reproduced without pain.

---

### Official Review · AnonReviewer3 · 2018-11-02
**Interesting method with solid results.**

**Rating:** 7
**Confidence:** 4

**Review:**

The authors propose to use a graph hypernetwork (GHN) to speedup architecture search. Specifically, the architecture is formulated as a directed acyclic graph, which will be encoded by the (bidirectional) GHN as a dense vector for performance prediction. The prediction from GHN is then used as a proxy of the final performance during random search. The authors empirically show that GHN + random search is not only efficient but also performs competitively against the state-of-the-art. Additional results also suggest predictions from GHN is well correlated with the ground truth obtained by the standard training procedure.

The paper is well-written and technically sound. While the overall workflow of hypernets + random search resembles that of SMASH (Brock et al., 2018), the architecture of GHN itself is a nontrivial and useful contribution. I particularly like the facts that (1) GHN seems flexible enough to handle richer topologies than many prior works (where each node in the graph is typically restricted to have a fixed number of neighbors), thanks to graphnets (2) the authors have provided convincing empirical evidence to back up their design choices about GHN through ablation studies.

In terms of experiments, perhaps one missing piece is to investigate alternative hypernet architectures in a controlled setting. For example, the authors could have implemented the tensor encoding scheme as in SMASH in their codebase to compare the capabilities of graph vs. non-graph structured hypernetworks.

I’m also curious about the stability of the algorithm and the confidence of the final results. What would be the standard deviation of the final performance if you repeat the entire experiments from scratch (training GHN+random search+architecture selection) using different random seeds?

A related question is whether the final performance can be improved with more compute. The algorithm is terminated at 0.84 GPU day, but I wonder how the performance would change if we keep searching for longer time (with more architecture samples). It would be very informative to see the curve of performance vs search cost.

---

> ### Author Response · Authors · 2018-11-27
> **Response to Reviewer 3**
>
> We thank the reviewer for their evaluation! To answer the questions:
>
> >> “I’m also curious about the stability of the algorithm and the confidence of the final results. What would be the standard deviation of the final performance if you repeat the entire experiments from scratch (training GHN+random search+architecture selection) using different random seeds?”
>
> We did not observe large variance when training the GHN on different seeds, and the variance for 10 architectures selected by the GHN is reported in Table 1.
>
> >> “A related question is whether the final performance can be improved with more compute. The algorithm is terminated at 0.84 GPU day, but I wonder how the performance would change if we keep searching for longer time (with more architecture samples). It would be very informative to see the curve of performance vs search cost.”
>
> Training was halted after the HyperNetwork showed convergence. We saw conducting the random search for longer lead to marginal improvements. Extending the random search to 4 GPU days gave 97.24 $\pm$ 0.05, compared to 97.16 $\pm$ 0.07 using 0.84 GPU days as reported. However, we suspect a more advanced search method would be able to utilize the additional compute time more efficiently.

---

### Public Comment · (anonymous) · 2018-10-09
**Interesting Work!**

This is a nice work. I have a few questions about the experiments.
(1) In Table 2, the search cost includes the training time and evaluation time on 1K models. Would you mind to also let us know the separate time of these two procedures?
(2) In Table 1 and Table 2, does (F=32) mean that the channels in the CIFAR architecture are 32(first conv)-32(6 cells)-64(6 cells)-128(6 cells)?
(3) In Sec.7.3, the hyper-parameters for optimization algorithms are different with some compared algorithms. These differences may lead a higher (or lower) accuracy. Would it be a little bit unfair?
(4) Do you plan to release the codes?
Thanks again, this is an interesting paper!

---

> ### Author Response · Authors · 2018-10-17
> **Thank you for your interest!**
>
> (1)
> For Table 2, the process takes 6 hours (GHN training) + 4 hours (15 sec/model evaluating) + 10 hours (retraining top 10 to select top 1). Note that 4 hours is an overestimate, as the code is not heavily optimized.
> For Table 1, there is no retraining phase.
>
> (2)
> Following (Liu et al., 2018c; Pham et al., 2018) , the first conv actually has 3x the number of channels F. So (F=32) means 96(first conv)-32(6 cells)-64(6 cells)-128(6 cells).
>
> (3)
> The hyperparameters for CIFAR-10 and ImageNet are chosen to match Liu et al., (2018c). One difference is the drop-path probability (0.4 vs 0.3). This was chosen ad hoc in earlier experiments and we did not observe a difference in results.
> For anytime, the hyperparameters are identical to Huang et al. (2018)
> Overall, we did not perform a grid search over hyperparameters. So it is possible that a grid search would improve our results.
>
> Perhaps the largest difference is that we accelerate training in a distributed fashion. However, our experiments showed negligible differences in accuracy compared to single GPU training.
>
> (4)
> We cannot say for certain at this moment, but we will consider releasing code after acceptance.
>
> Thanks again for your interest! Please let us know if any answers are unclear or if there are additional questions.

---

> > ### Public Comment · (anonymous) · 2018-10-17
> > **Most of my concerns are addressed.**
> >
> > Thanks for your reply. I still have two questions :)
> > 1. Table 3 misses the FLOPs for each method. Since we care about the computation cost on ImageNet, most previous methods show the FLOPs. I understand that the mobile set is < 600M FLOPs, but more precious FLOPs can always be helpful.
> > 2. Does "1x1 convolution" mean "ReLU-1x1Conv-BN (3 layers)"? Or a stack of two 1x1 Conv like the separable conv?
> >
> > Best Regards,

---

> > > ### Author Response · Authors · 2018-10-18
> > > **We are happy to answer any more questions!**
> > >
> > > Hi,
> > >
> > > 1. We will update the paper with the specific number of FLOPS.
> > > 2. 1x1 convolution means "ReLU-1x1Conv-BN"
> > >
> > > Thanks again!

---

> > > > ### Public Comment · (anonymous) · 2018-10-26
> > > > **Thanks for your reply.**
> > > >
> > > > Thanks for your reply.
> > > > May I ask the top-5 accuracy of "GHN Top-Best, 1K" in Table 3?

---

> > > > > ### Author Response · Authors · 2018-10-29
> > > > > **Thanks**
> > > > >
> > > > > The top-5 is 91.3
> > > > > We will also update the paper.
> > > > > EDIT: Corrected top-5 value.

---

### Author Response · Authors · 2018-11-27
**Overall response to reviewers**

We thank the reviewers for their comments. In addition to responding to the questions, we have updated the paper accordingly.

Regarding concerns around novelty, we agree that the idea of extending hypernetworks with graph neural networks is a natural one. However, as Reviewer 3 has mentioned, we argue that the design of GHN itself is nontrivial. We investigate how various aspects of the design impact the performance through extensive ablation studies. For example, we show the benefits of a novel forward-backward graph propagation scheme, and stacking GNNs in the depth dimension for parameter sharing on an architectural-motif level.

---

### Public Comment · (anonymous) · 2018-12-02
**Code availability?**

There was a previous question about code availability, and the response was:

"We cannot say for certain at this moment, but we will consider releasing code after acceptance."

I'd just like to emphasize that this would be very important for reproducibility, since there are a lot of moving pieces in this paper and I'm unsure whether the results can be reproduced without code. Also, to give a positive spin, if code was available I would expect a lot more excitement about this paper than otherwise (e.g., the DARTS paper led to a lot of excitement, in large part due to people being able to play with the code right away).

Therefore, I'd appreciate if the authors tried really hard to release their code. Thanks!

---

### Meta-Review · Area_Chair1 · 2018-12-13
**interesting contribution, competitive results**

**Confidence:** 5
**Recommendation:** Accept (Poster)

**Metareview:**

The paper proposes an architecture search method based on graph hypernetworks (GHN). The core idea is that given a candidate architecture, GHN predicts its weights (similar to SMASH), which allows for fast evaluation w/o training the architecture from scratch. Unlike SMASH, GHN can operate on an arbitrary directed acyclic graph. Architecture search using GHN is fast and achieves competitive performance. Overall, this is a relevant contribution backed up by solid experiments, and should be accepted.